# Metabolism and Chemical Degradation of New Antidiabetic Drugs (Part II): A Review of Analytical Approaches for Analysis of Gliptins

**DOI:** 10.3390/biomedicines11071956

**Published:** 2023-07-11

**Authors:** Anna Gumieniczek, Anna Berecka-Rycerz

**Affiliations:** Department of Medicinal Chemistry, Faculty of Pharmacy, Medical University of Lublin, Jaczewskiego 4, 20-090 Lublin, Poland; anna.berecka-rycerz@umlub.pl

**Keywords:** drug metabolism and drug degradation, chromatographic and radiometric methods, dedicated packings, quantitation and identification, gliptins

## Abstract

This paper is part II of the review on metabolism and chemical degradation of new antidiabetic drugs from glutides, gliflozins and gliptins. It is well known that metabolism data can be helpful for deriving safe levels of degradation impurities and their qualifying as far as toxicological aspects are concerned. As a result, it could link the quality of respective pharmaceutical products to clinical practice and patients. Some overlapping pathways of transformations of these important drugs of different chemical structures and different mechanisms of action were discussed. At the same time, the paper summarized interesting analytical tools for conducting modern drug metabolism as well as drug degradation experiments. The methods described here include liquid chromatography (LC) and liquid chromatography coupled with mass spectrometry (LC-MS or LC-MS/MS), which are widely used for detection and quantitative measurements of the drugs, their metabolites and degradants, as well as radiometric methods that are suitable for pharmacokinetic experiments. Special attention was paid to dedicated types of packing in chromatographic columns, as well as to special solutions in the LC-MS procedures. The present part addresses the analytical approaches elaborated for examining the metabolism and degradation pathways of gliptins that are dipeptidyl peptidase 4 (DPP-4) inhibitors.

## 1. Introduction

### Drug Metabolism and Drug Degradation Overlapping

Information on absorption, metabolism and excretion of drugs is necessary to support the studies on their pharmacokinetics and potential drug–drug interactions. Moreover, the knowledge on drug metabolism is one of the crucial factors used to assess their pharmacokinetic profile in patients with some dysfunctions. It is especially important in diabetic patients with higher incidence of chronic liver and kidney problems [1]. All of this creates the need to elaborate modern analytical methods suitable for determination of drugs and their metabolites, especially the drugs used for the treatment of diabetic patients [1,2]. On the other hand, drug stability studies are carried out to select the storage conditions and estimate the shelf life of pharmaceuticals. In addition, such studies allowed us to elaborate on the stability indicating methods to determine the drugs in the presence of their degradation products. Finally, such degradation studies are essential to inspect the changes in active pharmaceutical ingredients (APIs) and different drug products over the storage period and to assess the effect of environmental factors like pH, light and oxidative conditions on their chemical stability. Stability experiments in the stressed conditions aim to generate the maximum of degradation products in a short time in comparison with general stability studies (i.e., long term and accelerated stability studies). It is well known that some degradants may possess different biological properties than parent drugs, including toxic and immunity properties. Therefore, it is obligatory to estimate their levels in APIs or the marketed formulations, to know their structures, the way they are formed and finally, their potential amounts or concentrations [3].

The drug metabolism and drug degradation pathways may overlap, resulting in the formation of identical constituents. Therefore, the metabolism data can be helpful for deriving safe levels of degradation impurities and developing patient-centric therapy. Qualification of the degradation products or related substances is the process of acquiring and evaluating data to establish the biological safety of an individual degradation product, impurity or metabolite at the specified levels [4]. For all these purposes, information on metabolism of drugs from animal studies could be used to derive the no observed adverse effect level (NOAEL). Then, it could be converted to a human equivalent that was determined taking into account the safety and efficacy for patients. Additionally, such knowledge could be used to better understand the pharmaceutical product behavior on the market and more strongly connect the quality of this pharmaceutical product to clinical practice [5].

Generally, both drug metabolism in the body and drug degradation during processing and/or storage can undergo similar chemical reactions. Consequently, many impurities generated during degradation are also metabolites. It could be the important question when the process of qualification of impurities is concerned. It is obvious that simple comparison of an impurity with a detected metabolite in humans or animals is not adequate to qualify an impurity because the detected compound could be a metabolite or an impurity. Therefore, it is obligatory to establish that an exposure is equal to or greater than the level that might result from the real exposure to a particular impurity, following administration of some individual substance or product [5]. Thus, the ICH documents on respective thresholds at which impurities in new drug substances and products must be reported, identified and qualified must be taken into account [4].

Bearing in mind possible chemical connections between drug metabolism in the body and drug degradation under different environmental conditions, the goal of the present study was to collect the most interesting papers concerning three groups of important antidiabetics, i.e., gliptins (dipeptidyl peptidase 4 (DPP-4) inhibitors), glutides (glucagon-like peptide 1 (GLP-1) receptor agonists, GLP-1 RAs) and gliflozins (sodium glucose co-transporter 2 inhibitors, SGLT2 inhibitors), and to compare their metabolites and degradation products. We intended to find as many of the above connections as possible, as well as similar products of both metabolism and degradation, to express the idea of projecting the studies connecting these two essential aspects of antidiabetic therapy. The second goal of the present paper was to collect the most interesting analytical tools used to analyze these important drugs in terms of their metabolism and chemical degradation. Between them, liquid chromatography (LC) and liquid chromatography coupled with mass spectrometry (LC-MS), as well as high-resolution mass spectrometry (HRMS) were emphasized.

In Part I of our review (sent for publication in *Biomedicines*) we presented the data from the literature on metabolism, degradation and respective analytical tools for antidiabetic drugs from glutides (GLP-1 receptor agonists) and gliflozins (SGLT2 inhibitors) while the present Part II is devoted to gliptins (DPP-4 inhibitors).

## 2. Gliptins (DPP-4 Inhibitors)

Gliptins have been developed to prevent degradation of endogenously released incretins, glucose-dependent insulinotropic polypeptide (GIP) and glucagon-like peptide 1 (GLP-1), by DPP-4 enzyme. Thus, they act as DPP-4 inhibitors and prolong respective actions of these endogenous incretins. They do not pass the blood–brain barrier, have no direct effect on satiety, and in contrast with GLP-1 receptor agonists (GLP-1 RAs), did not alter gastric emptying. However, when compared with GLP-1 RAs, the DPP-4 inhibitors offer several advantages such as oral administration, absence of gastrointestinal adverse effects and lower costs. What is more, some clinical data suggest that gliptins could exert positive cardiovascular effects. Because of their safety profile, especially their very low risk of hypoglycemia, gliptins could be indicated in elderly patients [6,7].

Five of these DPP-4 inhibitors, i.e., anagliptin (ANA, N-[2-[[2-[(2S)-2-cyanopyrrolidin-1-yl]-2-oxoethyl]amino]-2-methylpropyl]-2-methylpyrazolo[1,5-a]pyrimidine-6-carboxamide) (approved in Japan), alogliptin (ALO, 2-{[6-(3-aminopiperidin-1-yl)-3-methyl-2,4-dioxo-3,4-dihydropyrimidin-1(2H)-yl]methyl}benzonitrile), linagliptin (LINA, 8-[(3R)-3-aminopiperidin-1-yl]-7-but-2-ynyl-3-methyl-1-[(4-methylquinazolin-2-yl)methyl]purine-2,6-dione), sitagliptin (SITA, (3S)-3-amino-1-[3-(trifluoromethyl)-5,6-dihydro[1,2,4]triazolo[4,3-a]pyrazin-7(8H)-yl]-4-(2,4,5-trifluorophenyl)butan-1-one), saxagliptin (SAXA, (1S,3S,5S)-2-[(2S)-2-amino-2-(3-hydroxyadamantan-1-yl)acetyl]-2-azabicyclo[3.1.0]hexane-3-carbonitrile) and vildagliptin (VILDA, (2S)-1-{[(3-hydroxyadamantan-1-yl)amino]acetyl}pyrrolidine-2-carbonitrile) (FDA approved), teneligliptin (TENE, [(2S,4S)-4-[4-(5-methyl-2-phenylpyrazol-3-yl)piperazin-1-yl]pyrrolidin-2-yl]-(1,3-thiazolidin-3-yl)methanone) (approved in Japan, South Korea and India [16) were introduced by regulatory authorities between 2006 and 2013, while evogliptin (EVO, (3R)-4-[(3R)-3-amino-4-(2,4,5-trifluorophenyl)butanoyl]-3-[(2-methylpropan-2-yl)oxymethyl]piperazin-2-one) was approved in South Korea in 2015. Omarigliptin (OMA, (2R,3S,5R)-2-(2,5-difluorophenyl)-5-[2-(methanesulfonyl)-2,6-dihydropyrrolo[3,4-c]pyrazol-5(4H)-yl]oxan-3-amine) and trelagliptin (TRELA, 2-({6-[(3R)-3-aminopiperidin-1-yl]-3-methyl-2,4-dioxo-3,4-dihydropyrimidin-1(2H)-yl}methyl)-4-fluorobenzonitrile) are the DPP-4 inhibitors approved in Japan in 2015 as the first once-weekly oral antidiabetic agents in the world [8].

Gliptins can be classified into peptidomimetic (i.e., ANA, OMA, SAXA, SITA, TENE and VILDA) and non-peptidomimetic (i.e., ALO, EVO, LINA and TRELA) subtypes. Due to their specificity to the substrate site of the enzyme, some of them have substituted pyrrolidines or thiazolidines as a proline mimetic moiety. Although their chemical structures differ from each other, all DPP-4 inhibitors are substrate-competitive active site binders and have common interactions with the key residues of the target protein [7]. Based on the half-life and time of dissociation from the DPP-4 enzyme, they are prescribed twice a day (e.g., ANA and VILDA), once a day (e.g., ALO, EVO, LINA, SITA, SAXA) or once a week (e.g., OMA, TRELA) [6]. The structures of the mentioned DPP-4 inhibitors are presented in Figure 1.

### 2.1. Metabolic Transformations of Gliptins (DPP-4 Inhibitors) and the Methods Used for Elucidating Their Metabolic Pathways

#### 2.1.1. Metabolism of Gliptins

Alogliptin (ALO) is a highly potent and selective inhibitor of DPP-4 that was developed using the technology of structure-based drug design (SbDD). In the paper from the literature [9], it was shown that ca. 10% of ALO is metabolized, while 60–70% of the dose is excreted as unchanged drug in the urine. Two minor metabolites were detected following oral administration of [14C]-ALO, i.e., N-demethylated ALO (ALO-M1) (<1% of the dose) (Table 1) that inhibits DPP-4 similar to the parent molecule, and inactive N-acetylated ALO (<6% of the dose). It was also speculated that ALO is mainly metabolized by CYP2D6 and CYP3A4.

Because anagliptin (ANA) is not frequently used in other countries than Japan, only a small number of reports exists investigating its biological properties and pharmacokinetics. The major metabolic pathway of ANA was proposed as the cyano group hydrolysis to generate the carboxylic acid metabolite ANA-M1 (Table 2), which accounted for 29.2% of the dose. The parent ANA and ANA-M1 were eliminated mainly with urine where the mechanisms of the active transport were probably involved [10].

More information on the possible metabolic pathways of evogliptin (EVO) in humans was reported. Metabolism of EVO was proposed as the phase I reactions where the drug is metabolized to 4-oxo-EVO (EVO-M1), 4(S)-hydroxy-EVO (EVO-M2) and 4(R)-hydroxy-EVO (EVO-M3) (Table 1), and as the II phase reactions forming 4(S)-hydroxy-EVO glucuronides and EVO-N-sulfates. At the same time, formation of EVO-M2 and EVO-M3 was inhibited by the CYP3A4 antibody, suggesting that CYP3A4 played a major role in the metabolism of EVO. It was also shown that EVO-M2 could be further metabolized to respective glucuronides by UDP-glucuronosyltransferases UGT2B4 and UGT2B7 [11]. 

The above data on EVO were confirmed in the next study from the literature [12]. The metabolism of EVO was also shown by the phase I reactions, i.e., hydroxylation and oxidation, as well as by opening of piperazine ring and oxidation, forming EVO-M4, EVO-M5 and EVO-M6 (Table 1). In addition, the phase II reactions were shown to be involved, i.e., glucuronidation, sulfation, and conjugation with glycine–cysteine moieties.

The chemical structure of linagliptin (LINA) is based on xanthine moiety that distinguishes this drug from other gliptins. It may offer some differences in pharmacokinetic and pharmacodynamic properties of LINA, e.g., its low dissociation from the enzyme and greater potency than other gliptins. It is known that LINA is predominantly eliminated unchanged after both oral and intravenous administration [7]. However, the inactive metabolite LINA-M1 was identified in plasma as a major metabolite after oral administration. A two-step mechanism was proposed for its formation, i.e., CYP3A4-dependent conversion of the secondary amine of the parent drug to the corresponding ketone via oxidative deamination, followed by reduction. In excreta, the main metabolite after oral and intravenous administration was LINA-M2 formed by hydroxylation of the methyl group of the butinyl side chain. Next, minor metabolites could be formed by combinations of the following reactions: oxidation of the butinyl side chain and the piperidine moiety followed by oxidative degradation of the piperidine moiety, and finally N-acetylation and glucuronidation. The oxidation of the methyl group at position 4 of the quinazoline moiety was also proposed, and resulted in the corresponding carboxylic acid metabolite LINA-M3 (Table 1). A cysteine adduct and its sulfate conjugate were additionally observed in urine after intravenous administration of LINA [13].

In the study of Xu et al. [14], absorption, metabolism and excretion of omarigliptin (OMA) were evaluated in healthy male subjects after a single oral dose of 25 mg of [14C]-OMA. Radioactivity levels in plasma and excreta were determined via accelerator mass spectrometry (AMS). As a result, minimal metabolism of OMA was observed, as indicated by the fact that the parent drug accounted for ca. 89% of the radioactivity in urine. However, some oxidative metabolites were detected in plasma, with each comprising less than 3% of the total radioactivity.biomedicines-11-01956-t001_Table 1Table 1Metabolites of DPP-4 inhibitors produced by the phase I reactions: alogliptin (ALO), anagliptin (ANA), evogliptin (EVO), linagliptin (LINA), saxagliptin (SAXA), sitagliptin (SITA), teneligliptin (TENE) and vildagliptin (VILDA).MetaboliteStructure*m*/*z* [M + H]^+^Ref.ALO-M1
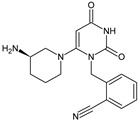
n.a.[9]ANA-M1
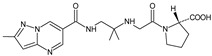
366 *[10]EVO-M1
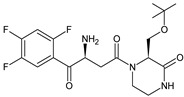
416 *[11]EVO-M2EVO-M3(isomers)
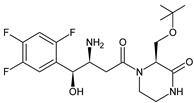
418 *[11,12]EVO-M4
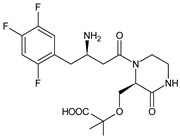
418 *[12]EVO-M5
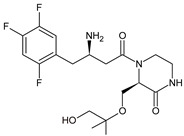
432 *[12]EVO-M6
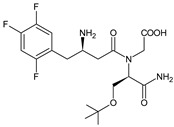
434 *[12]LINA-M1
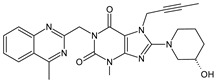
n.a.[7]LINA-M2
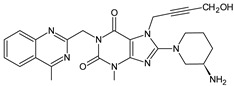
n.a.[13]LINA-M3
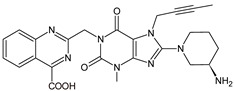
n.a.[13]SAXA-M1
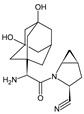
332 *[15,16,17]SITA-M1
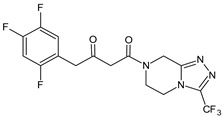
408 *[18]SITA-M2
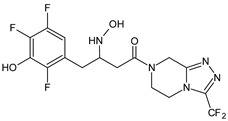
422 *[18]SITA-M3
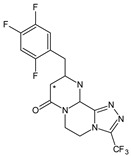
406 *[19]TENE-M1
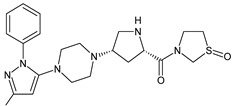
443 *[20,21]TENE-M2
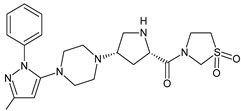
n.a.[20]TENE-M3
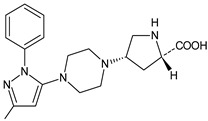
n.a.[20]TENE-M4
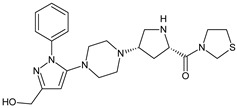
n.a.[20]TENE-M5
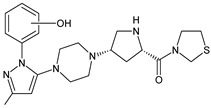
n.a.[20]VILDA-M1
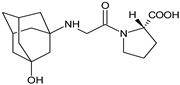
n.a.[7]n.a.—not available in the literature; *—low-resolution LC-MS.

A few metabolites of sitagliptin (SITA) were detected, e.g., diketone metabolite SITA-M1. In addition, hydroxylation of both amine group and aromatic ring followed by formation of glucuronide metabolites, as well as oxidation of NH2 and hydroxylation followed by loss of HF to form SITA-M2, was proposed (Table 1). On the other hand, it was clearly shown that only 3.1% of the parent drug was metabolized over 2 h incubation [19]. 

In the next paper concerning SITA, Vincent et al. [19] reported the phase I and phase II reactions of [14C]-SITA and detected its six metabolites in human plasma, urine and feces, using radiometric detection and LC-MS/MS, using the TurboIonSpray^®^ interface operated in positive ion mode. The metabolites were shown to be the products of hydroxylation and N-sulfation of SITA. In addition, oxidation of piperazine ring followed by cyclization was proposed to form SITA-M3 (Table 1). Glucuronidation was also included in SITA metabolism, but it was observed that glucuronides were further transformed and were not detected in feces.

Similar experiments elaborated for teneligliptin (TENE) indicated that 20–34% of the absorbed TENE was excreted through kidneys and 66–80% was metabolized. At the same time, multiple elimination pathways of TENE were proposed, which include metabolic reactions through CYP3A4 and flavin-containing monooxygenase 3 (CYP3A4/FMO3). The identified metabolites were described as TENE-M1 (thiazolidine-1-oxide), TENE-M2 (thiazolidine-1,1-dioxide), TENE-M3 (carboxylic acid formed by hydrolysis of the amide bond), TENE-M4 (hydroxymethyl metabolite formed by oxidation of the methyl group on pyrazole) and TENE-M5 (hydroxylated metabolite with the hydroxyl group on the phenyl moiety) (Table 1) [20]. 

As far as the next drug from gliptins, i.e., vildagliptin (VILDA) is concerned, its metabolism accounts for near 70% of the dose and is the dominant elimination pathway in humans. The major metabolite VILDA-M1 (Table 1) is pharmacologically inactive and is the product of hydrolysis of the cyano moiety. Next, identified metabolites of VILDA were formed by glucuronidation. At the same time, it was shown that the kidneys may be one of the major organs contributing to the hydrolysis of VILDA to VILDA-M1 [7].

In the next two studies from the literature [22,23], it was investigated whether different gliptins, i.e., ANA, ALO, LINA, SITA and VILDA, have the potential to covalently bind to macromolecules and, as a consequence, to initiate immune-mediated hepatotoxicity. As a result, the possibility of covalently binding to macromolecules was shown for ANA and VILDA. It was also found that VILDA and ANA, which both contain a cyanopyrrolidine moiety, rapidly reacted with L-cysteine in no enzymatic manner. Structural analysis of these adducts revealed that the nitrile moieties of both drugs were irreversibly converted to thiazoline acid. Thus, it was found that VILDA and ANA have the potential to covalently bind to a thiol residue of L-cysteine in proteins. Such binding may lead to unpredictable immune responses in humans.

#### 2.1.2. Analytical Methods for Elucidating the Metabolism of Gliptins

Structural characterization of ANA and its metabolites in humans was performed with radiometric and chromatographic methods (Table 2). Quantitative measurements in respective fractions of mobile phase were taken using a radio flow detector (for urine and feces) or liquid scintillation counter (LSC) (for plasma). Comparing the LC retention times and MS spectra with respective synthetic standards allowed identification of some of the detected metabolites [10].

To identify EVO and its metabolites in human liver microsomes, an orbitrap mass spectrometer coupled with the UPLC system was used. Separation was performed on a C8 column using gradient elution. A higher-energy collision dissociation (HCD) system was employed to investigate the fragmentation pattern of EVO and its metabolites (Table 2). As a result, five EVO metabolites were detected and identified via comparison with the retention times and MS/MS spectra of the corresponding standards [11]. In addition, a HPLC system coupled with a radioactivity detector (RAD) and an ion trap mass spectrometer (Table 2) was used in order to study of metabolism of [14C]-EVO after oral administration to male rats and dogs [12]. 

In the study of Xu et al. [14], absorption, metabolism and excretion of [14C]-OMA were evaluated in healthy male subjects via the AMS method. It is worth noting that AMS shows a special power to separate a rare isotope from an abundant neighboring mass [24]. In the mentioned study, the metabolite profiling was conducted using HPLC fractionation followed by AMS to generate radiochromatograms, and finally, metabolites were identified by LC-MS/MS with a hybrid system that consisted of a quadrupole and orthogonal acceleration TOF tandem mass spectrometer (Table 2). 

A specific and rapid LC-MS/MS method [15] was proposed for simultaneous determination of SAXA and its active metabolite, 5-hydroxy-SAXA (SAXA-M1) in human plasma. Sample preparation was accomplished via solid phase extraction (SPE) using SDS as an ion-pair reagent. In addition, next, the LC-MS/MS method for quantitative determination of SAXA and SAXA-M1 was described in the literature [16] (Table 2). 

In the study of Peng et al. [17], hydrophilic interaction liquid chromatography (HILIC) coupled with mass spectrometry (HILIC-MS) was applied to determine the plasma concentrations of SAXA and SAXA-M1 in the presence of metformin. Chromatographic separation was achieved on a HILIC Chrom Matrix HP amide column that was the optimal choice because of the high differences in hydrophilic properties of the mentioned analytes. Mass spectrometry was performed using a quadrupole tandem mass spectrometer in an MRM mode under positive ESI (Table 2). 

A similar analytical strategy was applied in the next study of Khreit et al. [18], where the metabolism of SITA was investigated via incubation of the drug with rat hepatocytes. For LC method, a zwitter ionic hydrophilic interaction ZIC-HILIC column with gradient elution and an orbitrap mass spectrometer were used (Table 2). 

Additionally, experiments on TENE metabolism were based on the use of AMS technology [20]. Metabolite profiling in plasma, urine and feces was performed via offline measurement of radioactivity using a HPLC system. Radioactivity was quantified using a LSC analyzer, with automatic quench correction by an external standard method, while identification of TENE and its metabolites was performed via ESI using a orbitrap hybrid Fourier Transform (FT) mass spectrometer (Table 2). Such hybrid mass spectrometer combines a linear ion trap MS and the orbitrap mass analyzer. Ions generated by a compound are collected and followed by axial ejection to the C-shaped storage trap which is used to store and cool ions before injection into the orbital trap. Signals from each of the orbital trap outer electrodes are amplified and transformed into a frequency spectrum by fast FT, which is finally converted into a mass spectrum [25]. 

Next, a rapid and sensitive LC-MS/MS method for simultaneous quantification of TENE and its active metabolite, TENE-M1, in human plasma was elaborated by Park et al. [21], using deuterated TENE as an internal standard. This method was proposed as an alternative to the AMS method described above [20], in order to avoid its disadvantages, including its complicated procedure and the expense of designing the radioactive isotope (Table 2). biomedicines-11-01956-t002_Table 2Table 2LC and LC-MS methods for determination of metabolites of gliptins: anagliptin (ANA), evogliptin (EVO), linagliptin (LINA), omarigliptin (OMA), saxagliptin (SAXA), sitagliptin (SITA), teneliglitin (TENE) and vildagliptin (VILDA).CompoundConditionsRef.ANAC18 column (2.0 × 150 mm, 3 μm) and isocratic elution using 1% CH_3_COOH/ACN (80:20, *v*/*v*), 1.0 mL/min;MS/MS with positive ESI.C18 column (4.6 × 250 mm, 5 μm) and gradient elution: (A) 50 mM ammonium acetate, (B) ACN, 1.0 mL/min.[10]EVOUnison-C8 column (2.0 × 75 mm, 3.0 μm) and gradient elution of A) 5% ACN in 0.1% HCOOH and B) 95% ACN in 0.1% HCOOH, 0.3 mL/min. The column and autosampler temperatures: 40 °C and 6 °C; ESI in positive and negative mode: spray voltage, 4.0 kV in positive mode and −3.0 kV in negative mode; vaporizer temperature, 350 °C; capillary temperature, 330 °C; sheath gas pressure, 35 Arb; and auxiliary gas pressure, 15 Arb; collision energy of 10 to 40 eV.[11]EVOKinetex C18 column (4.6 × 150 mm, 2.6 μm) and gradient elution with (A) 20 mM ammonium acetate (pH 4) and (B) ACN, 1 mL/min. The column temperature 40 °C and UV detection at 268 nm.ESI in the positive mode: spray voltage, 4.5 kV; vaporizer temperature, 350 °C; capillary temperature, 330 °C; sheath gas pressure, 50 Arb; auxiliary gas pressure, 10 Arb; and sweep gas pressure, 5 Arb, the collision gas He, and the normalized collision energy during product ion scanning 35%.[12]OMAACE 5 C8 column (4.6 × 250 mm; 5 µm) and gradient elution with (A) 2 mM ammonium acetate in ACN:H_2_O (5:95) containing 0.1% HCOOH, and (B) 2 mM ammonium acetate in ACN:H_2_O (95:5) containing 0.1% HCOOH, 1.0 mL/min; N_2_ as the nebulizer and auxiliary gas, and Ar as the collision gas. The ESI capillary voltage 1.2 kV. The source and desolvation temperatures 100 and 550 °C; collision energy from 20 to 30 eV.[14]SAXAACE CN column (4.6 × 150 mm, 5 μm) and ACN and 10.0 mM ammonium formate buffer of pH 5.0 (80:20, *v*/*v*); triple quadrupole MS detection with positive ESI.[15]SAXAC18 column (2.1 × 50 mm, 5 µm) and gradient elution with (A) 0.1% HCOOH in H_2_O and (B) 0.1% HCOOH in ACN. TurboIonSpray^®^ source, positive ionization mode, using SRM. N_2_ as the nebulizer, curtain and collision gas; 450 °C for TIS interface, 5000 V setting for ion spray voltage, 30 setting for the curtain gas and nebulizer gas.[16]SAXAHILIC Chrom Matrix HP amide column (3.0 × 100 mm, 5 μm), ACN and 5 mM ammonium formate buffer containing 0.1% HCOOH. Ion spray voltage 5500 V, ion spray temperature 550 °C, ion source gas 1: 50 Arb, ion source gas 2: 55 Arb, curtain gas (N_2_) 30 Arb, collision gas (N_2_) 10 Arb, entrance potential 10 V, collision cell exit potential 12 V.[17]SITAZIC-HILIC column (4.6 mm × 150 mm, 5 µm) and gradient elution with (A) HCOOH in H_2_O (0.1% *v*/*v*) and (B) HCOOH (0.1% *v*/*v*) in ACN, 0.3 mL/min.The capillary temperature 250 °C, spray voltage +4.5 kV and the sheath and auxiliary gas (N_2_) flow rates 45 and 15. CID voltage of 40 eV.[18]TENECAPCELL PAK C18 UG120 column (4.6 × 250 mm, 5 µm,) at 40 °C and gradient elution with (A) 20 mM ammonium acetate and (B) ACN, 0.8 mL/min; positive ESI at 3800 V and CID at the collision energy of 35%.[20]TENECAPCELL Pak C18 column (2.0 × 2150 mm, 5 μm) and isocratic elution with ACN, MeOH and H_2_O, 025 mL/min; temperature of the column and autosampler 50 °C and 10 °C. MS: collision gas 5 psi, curtain gas 10 psi, ion source gas (nebulizer) 30 psi, ion spray voltage 5500 V, and collision energy of 37 eV for TENE; declustering potential, entrance potential, and collision exit potential were 106 V; ESI positive ion mode using MRM.[21]ANAALOLINASITAVILDAUPLC system and a QQQ mass spectrometer equipped with a switching valve;XBridge C18 column (2.1 × 50 mm, 3.5 μm) and gradient with (A) 0.5 mM ammonium hydrogen carbonate and (B) MeOH for VILDA or (A) 1 mM ammonium acetate and (B) ACN (B) for ANA, ALO, SITA and LINA; 0.55 mL/min. The autosampler 4 °C. ESI positive ion mode using MRM transitions.Orbitrap Fusion MS system coupled with a HPLC system: XBridge C18 column (4.6 × 100 mm, 5 μm) and gradient elution with (A) 20 mM ammonium acetate (A) and (B) ACN (B), 1.0 mL/min; ESI positive and negative ion mode.[22,23]Arb—arbitrary units.

### 2.2. Chemical Degradation of Gliptins (DPP-4 Inhibitors) and the Methods Used to Elucidate Their Degradation Pathways

#### 2.2.1. Chemical Degradation of Gliptins

Using HPLC method with two detectors, i.e., UV and charged aerosol detector (CAD), it was shown that alogliptin (ALO) degraded about 60% in different stress conditions. After acidic degradation, two additive peaks were observed with both the CAD and UV detectors. One peak was observed early in the run and the other after the peak of the parent ALO. At the same time, the retention times observed with the CAD and the UV detector were not the same, suggesting the presence of two different compounds. Under thermal conditions, one degradation peak was observed in both detectors after the peak of parent ALO, with a retention time similar to the peak observed in the UV detector in acidic conditions. Under the photolytic conditions, one degradation peak before ALO using the CAD detector and one peak after ALO using the UV detector were observed, similarly to the peaks detected after acidic degradation [26]. 

Other work from the literature showed significant degradation of ALO leading to five degradants, i.e., ALO-D1 in acidic, ALO-D2 and ALO-D3 in alkaline, ALO-D4 in thermal, and ALO-D3 and ALO-D5 in oxidative conditions (Table 3). ALO-D1 was the major degradation product under acidic stress with increased chromatographic retention when compared to ALO, suggesting that its polarity was weaker than that of the parent drug. Based on the results of MS analysis, the structure with the chemical name of 2-[[3,4,5,6-tetrahydro-3-methyl-2,4,6-trioxo-1(2H)-pyrimidinyl]methyl]-benzonitrile was proposed. The ALO-D2 product was one of the major alkaline degradants with the shortest retention among all detectable related substances. Therefore, its polarity should be relatively high in comparison with the parent ALO. Based on the results of MS analysis, ALO-D2 was assigned as 2-aminomethylbenzonitrile, resulting from the hydrolysis of 3,4-dihydro-3-methyl-2,4-dioxo-1(2H)-pyrimidinyl ring. The next degradant ALO-D3 was the major degradation product under both alkaline and oxidative stress, with reduced retention under the reverse phase conditions in comparison with ALO, but stronger in comparison with ALO-D2. This product was identified as 2-[[6-(3-amino-1-piperidinyl)-3,4-dihydro-3-methyl-2,4-dioxo-1(2H)-pyrimidinyl]methyl]-benzamide. The ALO-D4 was the major degradation product in thermal stress, with polarity similar to that of ALO. Based on the data obtained, its structure was proposed as 2-[[6-(3-piperidinylamino)-3,4-dihydro-3-methyl-2,4-dioxo-1(2H)-pyrimidinyl]methyl]benzonitrile. Next, ALO-D5 was one of the major degradation products under oxidative stress with increased retention in comparison with the parent ALO. Considering oxidation of C=C bond in 3,4-dihydro-3-methyl-2,4-dioxo-1(2H)-pyrimidinyl ring and its weaker polarity than ALO, it could be concluded that oxidative substitution can occur in position 5 of the pyrimidinyl ring. Thus, ALO-D5 could be identified as 2-[[6-(3-amino-1-piperidinyl)-3,4-dihydro-5-hydroxy-3-methyl-2,4-dioxo-1(2H)-pyrimidinyl]methyl]benzonitrile (Table 3) [27].

The next few studies were assessed to examine the degradation behavior of anagliptin (ANA) under different stress conditions followed by elucidation of the structures of respective degradants. Overall, seven degradants (from ANA-D1 to ANA-D7) were identified (Table 3) [28,29]. Formation of ANA-D1 and ANA-D2 can be explained by hydrolysis of amide bond adjacent to 2-methylpyrazolo[1,5-a]pyrimidine ring and pyrrolidine-2-carbonitrile. Next, the nitrile group attached to the pyrrolidine ring is converted to carboxylic acid and amide to form ANA-D3 and ANA-D4, respectively. It is worth noting that ANA-D3 was structurally similar to the ANA-M1 metabolite (Table 1). Interestingly, Thorpe-Ingold effect or gem-dimethyl effect was observed in the formation of ANA-D5 and ANA-D7. According to this, ring cyclization is enhancing with the increase in the size of two substituents on a tetrahedral center [30]. 

Based on this gem-dimethyl effect, formation of ANA-D5 can be explained by hydrolysis of nitrile to amide followed by cyclization between the imine carbon and secondary amine group with loss of the water molecule. Similarly, ANA-D7 could be formed via hydroxylation of methylpyrazolopyrimidine ring followed by cyclization with an elimination of water molecule. As far as ANA-D6 was concerned, it was formed due to N-hydroxylation of the secondary amine (attached to tetrahedral carbon) under oxidative stress conditions. Generally, ANA degraded under acidic (ANA-D1, ANA-D2, ANA-D3, ANA-D5 and ANA-D5), basic (ANA-D4), neutral hydrolytic (ANA-D2, ANA-D4 and ANA-D5) and oxidative (ANA-D3, ANA-D4, ANA-D6 and ANA-D7) stress conditions (Table 3) [28,29]. In the study of Pandya et al. [31], two degradation products were detected, i.e., ANA-D4 that was described above [28,29] and one new degradant ANA-D8 (Table 3). This oxidative degradant was characterized as N-[2-({2-[(2S)-2-cyanopyrrolidin-1-yl]-2-oxoethyl}amino)-2-methylpropyl]-2-methylpyrazolo-[1,5-a]pyrimidine-N-oxido-6-carboxamide [31].

Upon forced degradation, 12 related substances of linagliptin (LINA) were obtained (Table 3). Under basic hydrolysis, addition of the N-((4-methylquinazolin-2-yl)methyl)formamide group of LINA to its free amino group leading to LINA-D1, i.e., (R)-1-(1-(7-(but-2-ynyl)-3-methyl-1-((4-methylquinazolin-2-yl)methyl)-2,6-dioxo-2,3,6,7-tetrahydro-1H-purin-8-yl)piperidin-3-yl)-3-((4-methylquinazolin-2-yl)methyl)urea was proposed. Under acidic hydrolysis, LINA-D2, LINA-D3 and LINA-D4 were detected. Between these three degradants, formation of LINA-D2 was proposed due to hydrolysis of the quinazoline structure leading to (R)-N-(2-acetylphenyl)-2-(8-(3-aminopiperidin-1-yl)-7-(but-2-ynyl)-3-methyl-2,6-dioxo-2,3,6,7-tetrahydro-1H-purin-1-yl)acetamide. The other two acid hydrolysis products (two isomers, LINA-D3 and LINA-D4) were proposed as 8-(3-aminopiperidin-1-yl)-7-(3-chlorobut-2-enyl)-3-methyl-1-((4-methylquinazolin-2-yl)methyl)-1H-purine-2,6(3H,7H)-dione, formed by addition of HCl to the alkyne side chain of the parent drug. Under oxidative stress, the next degradants (from LINA-D5 to LINA-D8) were formed. LINA-D7 and LINA-D8 produced similar spectra and they were assigned as isomers of 1-(7-(but-2-yn-1-yl)-3-methyl-1-((4-methylquinazolin-2-yl)methyl)-2,6-dioxo-2,3,6,7-tetrahydro-1H-purin-8-yl)-3-iminopiperidine 1-oxide. Their formation was proposed as a result of N-oxidation, as well as oxidation of the amine of LINA to an imine. The structures of the next two degradants, i.e., LINA-D9 and LINA-D10, were assigned as 7-(but-2-ynyl)-8-(3-hydroperoxypiperidin-1-yl)-3-methyl-1-((4-methylquinazolin-2-yl) methyl)-1H-purine-2,6(3H,7H)-dione and 7-(but-2-ynyl)-3-methyl-1-((4-methylquinazolin-2-yl) methyl)-8-(3-nitropiperidin-1-yl)-1H-purine-2,6(3H,7H)-dione, respectively. The first was formed due to oxidation of the free amino group of LINA, while the last was the result of oxidation of the free amino group of LINA to a nitro group (Table 3) [32]. 

The next paper from the literature showed above 20% degradation of LINA in acidic conditions and the presence of one new degradation product, i.e., LINA-D11 (Table 3). Based on the spectral data, the structure of this degradation product was characterized as (R)-N-(2-acetylphenyl)-2-(8-(3-aminopiperidin-1-yl)-7-(but-2-yn-1-yl)-3-methyl-2,6-dioxo-2,3,6,7-tetrahydro-1H-purin-1-yl)acetamide. The mechanism of formation of this impurity was proposed as nucleophilic addition of H2O through the catalysis of hydrogen proton to the imine group in quinazoline and further hydrolysis of this new imine group. What is more, it was emphasized that this impurity contains the N-acylated amino aryl groups, which are well-known DNA reactive substances [33]. 

Forced degradation studies were also conducted for tablets containing LINA and metformin. As a result, three new possible degradants were identified, i.e., LINA-D13, LINA-D14 and LINA-D15 (Table 3). The LINA-D13 was formed due to the presence of reducing sugar, while the LINA-D14 was formed due to the presence of HCOOH (because of their presence as tablet excipients). As far as LINA-D15 was concerned, it was formed in the presence of metformin in these two component tablets. However, the authors also proposed a general mechanism to form amide in the presence of amine and acid that could also be adequate for other similar APIs [34]. 

It is also worth noting that some studies from the literature concerning LINA and its process-related impurities showed possible risk of toxicity of these molecules, according to different computational programs, e.g., pKCSM, Osiris and LAZAR [35]. It was concluded that some impurities of LINA presented prediction of mutagenicity with the use of all software used. Moreover, some suggestions of carcinogenic effect with the use of Osiris and LAZAR were discussed. What is more, an experimental Comet assay was performed, in which LINA and its process related impurities showed some DNA damage at concentrations of 10 × Cmax, which was in accordance with the suggestions for in silico studies mentioned above. 

The work of Tantawy et al. [36] was elaborated for profiling the chemical stability of omarigliptin (OMA). The formed degradation products were then identified by means of IR and MS. It was found that OMA showed stability under photolytic and thermal conditions, but was sensitive to acidic and alkaline conditions, where complete degradation was achieved, and similar degradants were observed. This assumption was confirmed by LC-MS analysis, where their positive mass scan determinations showed the same molecular ion peaks. OMR was also susceptible to oxidative degradation and was completely oxidized when stored at room temperature for 50 h. The formation of the oxidative derivative product OMA-D1 (Table 3) was confirmed via the respective GC-MS method.

In the study of Sridhar et al. [37], comprehensive estimation of the chemical stability of saxagliptin (SAXA) was performed. Acidic and basic stress resulted in four degradants of SAXA, from SAXA-D1 to SAXA-D4, while oxidative conditions led to the occurrence of SAXA-D1 and SAXA-D4, as well as three more degradants, from SAXA-D5 to SAXA-D7 (Table 3). The spectrum of SAXA-D1 displayed the characteristics that suggested the loss of a carbonyl group and ammonia; however, the ions due to adamantyl and amine groups were still present in the molecule. Finally, it was confirmed as 4-((1r,3s,5R,7S)-3-hydroxyadamantan-1-yl)-6-iminohexahydro-1H-cyclopropa[4,5]pyrrolo[1,2-a]pyrazin-3(1aH)one. Additionally, the spectrum of SAXA-D2 indicated the presence of an adamantyl ring. Based on the respective fragmentation pattern, its structure was confirmed as 2-amino-2-(3-hydroxy-1-adamantyl)acetyl]-2-azabicyclo[3.1.0]-hexane-3-carboxylic acid. The product ions for SAXA-D3 confirmed that the carboxylic group was connected to the methylene carbon attached to the adamantyl ring. Thus, this degradant was described as {(2-azabicyclo[3.1.0]hex-3-yl)carboximidoylamino}(3-hydroxy-1-adamantanyl)acetic acid. Finally, SAXA-D4 was confirmed as 2-[2-amino-2-(3-hydroxy-1-adamantanyl)acetyl]-2-azabicyclo[3.1.0]hexane-3-carboxamide, SAXA-D5 as a primary N-oxide of 2-[2-amino-2-(3-hydroxy-1-adamantanyl)acetyl]-2-azabicyclo[3.1.0]hexane-3-carbonitrile while SAXA-D6 was confirmed as monohydroxy-substituted SAXA, with an -OH group added to the adamantyl moiety.

In the study from the literature concerning sitagliptin (SITA) [38], SITA-D1 and SITA-D2 degradants were detected after acidic, alkaline and thermal degradation. The molecule of SITA-D1 suggested that the triazolopyrazine fragment of the parent SITA was released from the structure. Next, the authors suggested that SITA could cleave between the N7 of triazolopyrazine moiety and the carbonyl group to form SITA-D2. Thus, the abovementioned degradant could be identified as (R)-3-amino-4-(2,4,5-trifluorophenyl)butanoic acid and 3-(trifluoromethyl)-5,6,7,8-tetrahydro[1,2,4]triazolo[4,3-a]pyrazine, respectively. Next, the triazolopyrazine structure could be oxidized to form SITA-D3 (Table 3). 

What is more, the next two degradants, i.e., SITA-D4 and SITA-D5 were identified after oxidative degradation of SITA [39]. Their exact masses indicated that they were formed by dehydration of 3-aminobutan-1-one. The results showed that they could be positional isomers with respect to the double bond, where both molecules had the same *m*/*z* (Table 3). 

In the next study from the literature [40], SITA-D1, SITA-D2 and SITA-D3 degradants were identified after degradation in acidic and alkaline conditions. In addition, two new products with the same *m*/*z*, i.e., SITA-D7 and SITA-D8 were detected after decomposition of the parent drug with the loss of ammonia from an abundant ion under basic and oxidative stress (Table 3).

Teneligliptin (TENE) was stressed under basic, oxidative and thermal conditions, and showed several degradation products, i.e., TENE-D1 ((4-(4-(1-aminovinyl)piperazin-1yl)pyrrolidin-2-yl)(thiazolidin-3-yl)methanone), TENE-D2 (N,N-diethyl-1-phenyl-1H-pyrazol-5-amine) and TENE-D3 (1-(pyrrolidin-3-yl)piperazine) that were observed in both basic and thermal conditions. Next, characteristic products TENE-D4 (4-(4-(1-ethyl-3-methyl-1H-pyrazol-5-yl)piperazin-1-yl)-N-(mercaptomethyl)-N-methylpyrrolidine-2-carboxamide) and TENE-D5 ((4-(4-(3-methyl-1-vinyl-1H-pyrazol-5-yl)piperazin-1-yl)pyrrolidin-2-yl)(thiazolidin-3-yl)methanone) were observed in basic or thermal conditions, respectively. It was interesting to observe that the products formed with photolytic stress were completely different than the abovementioned degradants. Their structures were described as N,N-diethyl-1Hpyrazol-5-amine (TENE-D6) and 2-amino-N-(mercaptomethyl)-N-methylacetamide (TENE-D7) (Table 3) [41].

When experiments on stress degradation of trelagliptin (TRELA) were conducted, the acidic and oxidative degradation product of TRELA was found to be 2-[(3-methyl-2,4,6-oxo-tetrahydro-pyrimidin-1(2H)yl)-methyl]-4-fluorobenzonitrile (TRELA-D1). What is more, TRELA-D2, which is similar to the previously described ALO-D3, was identified under oxidative degradation (Table 3). This could be due to the structural similarity between TRELA and ALO except with a less fluorine atom [27,42,43]. 

In the study of Luo et al. [44], a rapid, sensitive and accurate HPLC-UV method was developed for separation of TRELA and its eight potential process-related impurities. Between many intermediates and byproducts, three degradants were proposed, i.e., TRELA-D1 and TRELA-D2, which were mentioned above, and one new degradation product TRELA-D3 (Table 3). In the study of Zhang et al. [45] TRELA-D2 was identified as (R)-2-[6-(3-amino-piperidin-1-yl)-3-methyl-2,4-dioxo-3,4-dihydro-2H-pyrimidin-1-ylmethyl]-4-fluoro-bezamide. When TRELA was exposed to oxidation, TRELA-D2 and one new degradant, i.e., TRELA-D4, were found. It was proposed that TRELA-D4 might be (R)−2-[6-(3-amino-piperidin-1-yl)-3-methyl-2,4-dioxo-3,4-dihydro-2H-pyrimidin-1-ylmethyl]-4-fluoro-benzoic acid. It has one additional carboxylic acid group but is missing one cyano group and succinic acid compared to that of the parent TRELA (Table 3). 

Forced degradation of VILDA was performed at extreme pH values and in oxidative conditions [46]. UHPLC-DAD-MS experiments allowed us to separate and identify four degradants of VILDA, from VILDA-D1 to VILDA-D4 (Table 3). VILDA-D1 was detected when the drug was stressed in acidic and basic conditions. It was supposed that the pyrrolidine-2-carbonitrile motif was left from the structure of the parent drug to produce the corresponding carboxylate. Thus, it could be described as [(3-hydroxytricyclo[3.3.1.13,7]decan-1-yl)amino]acetic acid. As far as VILDA-D2 is concerned, it was detected in basic, oxidative and acidic conditions. It was probably formed via hydrolysis of the cyano group of VILDA into the amide one. It could be identified as 1-{[(3-hydroxytricyclo[3.3.1.13,7]decan-1-yl)amino]acetyl}pyrrolidine-2-carboxamide. Thus, the results from the literature pointed to hydrolysis of the cyano group in basic, oxidative, as well as acidic conditions. Further hydrolysis of the amide group of VILDA-D2 afforded the corresponding carboxylic acid, leading to VILDA-D3 (Table 3). 

In the next study from the literature [47] separation of VILDA and its next two impurities, i.e., 2-pyrrolidinecarboxamide (VILDA-D4) and 3-amino-1-adamantanol (VILDA-D5) was achieved (Table 3). In addition, the next six degradants of VILDA were reported by Arar et al. [48]. One degradant was formed under acidic conditions and was described as 2-((1R,3S,5R,7S)-3-hydroxyadamantan-1-yl)hexahydropyrrolo[1,2-a]pyrazine-1,4-dione (VILDA-D6). Two degradants were formed under basic hydrolysis (VILDA-D7 and VILDA-D8). Some new degradants were formed under oxidative stress. One of them was designated as N-hydroxy-N-((1R,3S,5R,7S)-3-hydroxyadamantan-1-yl) glycinate (VILDA-D9) (Table 3). biomedicines-11-01956-t003_Table 3Table 3Stress degradation products of DPP-4 inhibitors: alogliptin (ALO), anagliptin (ANA), linagliptin (LINA), omarigliptin (OMA), saxagliptin (SAXA), sitagliptin (SITA), teneligliptin (TENE), trelagliptin (TRELA) and vildagliptin (VILDA).Stress ConditionsDegradantStructure*m*/*z*[M + H]+Ref.[H^+^]ALO-D1
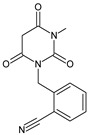
258 **[27][OH^−^]ALO-D2
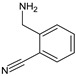
133 **[27][OH^−^][O]ALO-D3
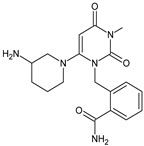
358 **[27][T]ALO-D4
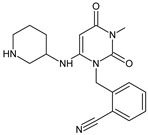
340 **[27][O]ALO-D5
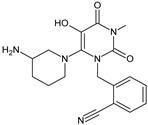
356 **[27][H^+^]ANA-D1
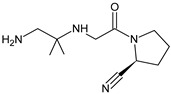
225 *[28][H^+^]ANA-D2
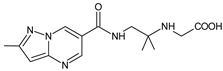
306 *[28,29][H^+^][O]ANA-D3
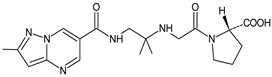
403 *[28,29][H^+^][OH^−^][O]ANA-D4
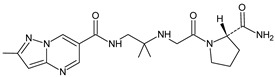
402 *[28,29,31][H^+^]ANA-D5
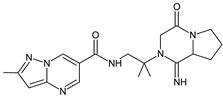
384 *[28][O]ANA-D6
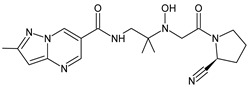
400 *[28][O]ANA-D7
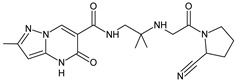
400 *[28][O]ANA-D8
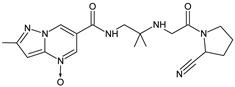
398 *[31][OH^−^]LINA-D1
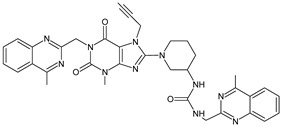
672 **[32][H^+^]LINA-D2
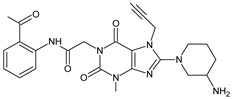
492 **[32][H^+^]LINA-D3LINA-D4(isomers)
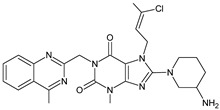
509 **[32][O]LINA-D5
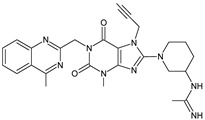
514 **[32][O]LINA-D6
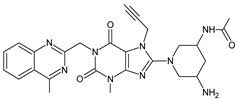
530 **[32][O]LINA-D7LINA-D8(isomers)
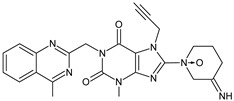
487 **[32][O]LINA-D9
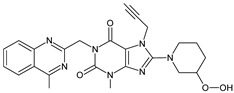
490 **[32]
LINA-D10
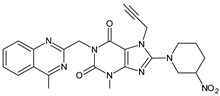
503 **[32]
LINA-D11
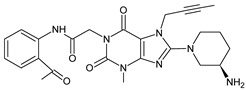
492 **[32]
LINA-D12
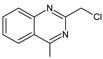
n.a.[33]reducing sugarLINA-D13
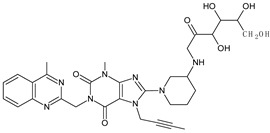
635 **[34]HCOOHLINA-D14
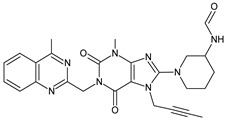
501 **[34]metforminLINA-D15
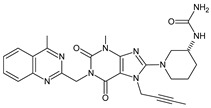
516 **[34][O]OMA-D1
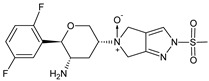
415 *[36][H^+^][OH^−^]SAXA-D1
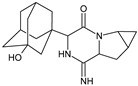
316 **[37][H^+^][OH^−^]SAXA-D2
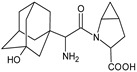
335 **[37][H^+^][OH^−^]SAXA-D3
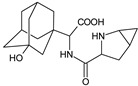
335 **[37][H^+^][OH^−^]SAXA-D4
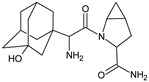
334 **[37][O]SAXA-D5
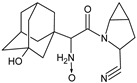
332 **[37][O]SAXA-D6
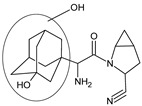
332 **[37][O]SAXA-D7
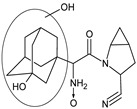
348 **[37][H^+^][OH^−^][T]SITA-D1
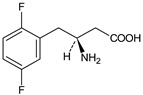
234 *[38,39,40][H^+^][OH^−^][T]SITA-D2
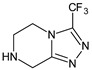
193 *[38,39,40][H^+^][OH][O]SITA-D3
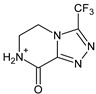
207 *[39,40][O]SITA-D4SITA-D5
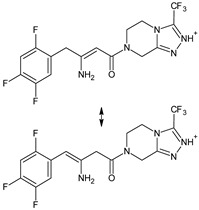
406 *[39][OH^−^][O]SITA-D6SITA-D7
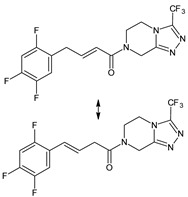
391 *[40][OH^−^][T]TENE-D1
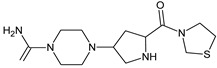
310 *[41][OH^−^][T]TENE-D2
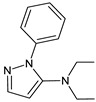
214 *[41][OH^−^][T]TENE-D3
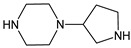
156 *[41][OH^−^]TENE-D4
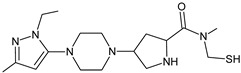
355 *[41][T]TENE-D5
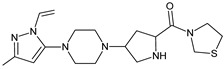
376 *[41][O]TENE-D6
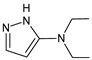
139 *[41][O]TENE-D7
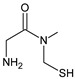
137 *[41][H^+^][O]TRELA-D1
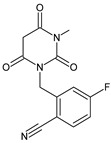
276 **[43,44,45][OH^−^][O]TRELA-D2
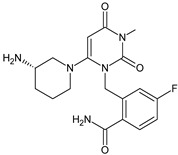
376 **[44,45][H^+^][O]TRELA-D3
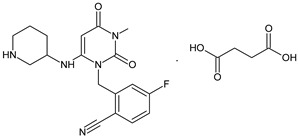
358 **[44][O]TRELA-D4
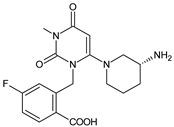
378 *[45][H^+^][OH^−^]VILDA-D1
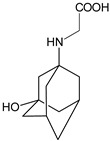
226 *[46][H^+^][OH^−^][O]VILDA-D2
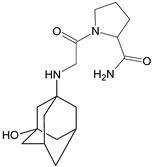
322 *[46][H^+^][OH^−^]VILDA-D3
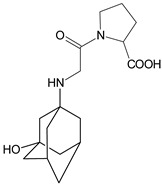
323 *[46][OH^−^]VILDA-D4
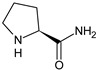
115 **[47][OH^−^]VILDA-D5
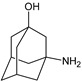
168 **[47][H^+^]VILDA-D6
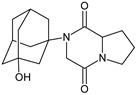
304 *[48][OH^−^]VILDA-D7
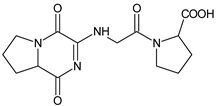
323 *[48][OH^−^]VILDA-D8
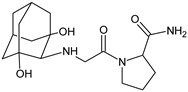
337 *[48][O]VILDA-D9
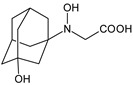
241 *[48]n.a.—not available in the literature; *—low-resolution LC-MS; **—high-resolution LC-MS.

#### 2.2.2. Methods for Elucidating Degradation Pathways of Gliptins

Stability-indicating HPLC methods using an UV detector and a charged aerosol detector (CAD) were elaborated for the assessment of chemical stability of ALO. The analysis was performed on a C8 column (4.6 × 250 mm, 5 μm) at a flow rate of the mobile phase of 0.8 mL/min, using ACN-10 mM acetate buffer of pH 3.5 (90:10, *v*/*v*) and UV detection at 275 nm [26]. The above method is worth mentioning because of its use of the CAD detector, which may offer entirely new possibilities by exploiting a large range of analytes. This is frequently impossible when using other detection methods. Other benefits of using CAD include an accurate response for all nonvolatile compounds independently of the analyte structure and high sensitivity [49].

Other work from the literature [27] showed significant degradation of ALO leading to five degradants. Highly specific and efficient LC-MS-Q-TOF was used for their separation. The eluent was split and introduced into the ESI source of the Q-TOF unit, operating in positive ionization mode (Table 4).

The next few studies from the literature were assessed to examine the degradation behavior of ANA under different stress conditions followed by elucidation of the structures of respective degradants. All the stressed samples were analyzed using UPLC/PDA and LC-MS/MS methods (Table 4) [28,29]. In the study of Pandya et al. [31], preparative HPLC separation of ANA and its degradants was performed with a chromatographic system with a PDA detector and a Daisogel-SP-100–10-ODS-P column (20 × 250 mm, 10 μm). The flow rate of the mobile phase was kept at 5 mL/min. A HRMS system consisting of an UHPLC, NANO Chipcube^®^ interface and iFunnel Q-TOFs was used (Table 4).

Separation of LINA and its 12 stress degradation products was achieved using HPLC-PDA and LC-MS/MS experiments [32] (Table 4). Separation of LINA and its degradation products was also achieved on a Zorbax Eclipse XDB C18 column (250 × 50 mm, 10 μm) using mobile phases consisting of phosphate buffer (20 mM) of pH 2.5) and ACN with the flow rate of 0.6 mL/min. The gradient program, the column temperature 30 °C and UV detection at 225 nm were used [33].

Forced degradation studies were also conducted for tablets containing LINA and metformin [34]. Separation of the drugs and degradants was achieved using LC-MS and UPLC-TOF/MS methods with a Zorbax SB-Aq column, which is a RP HPLC phase which is designed to retain hydrophilic and other compounds when using highly aqueous mobile phases, even including 100% water (Table 4).

The work of Tantawy et al. [36] was elaborated for profiling of chemical stability of OMA under various stress conditions. The formed degradation products were then identified by means of IR and MS. In addition, thin-layer chromatography (TLC) with mobile phase of MeOH:ethyl acetate:33% NH4OH (2:8:1, *v*/*v*/*v*) was used for separation. What is more, formation of the oxidative degradant of OMA (OMA-D1) was confirmed by the respective GC-MS method.

In the study of Sridhar et al. [37], comprehensive estimation of chemical stability of SAXA was performed using LC-PDA, LC-MS, and LC-MS/MS methods. In the next study [40], a new LC-MS analysis of SITA and its degradants was performed using an accurate mass Q-TOF spectrometer equipped with a dual ESI-Jet stream ion source (Table 4).

To develop a suitable RP-HPLC method for identifying TENE and its degradation products, a system with a variable wavelength UV detector was used [41]. TENE was stressed under different conditions and the obtained samples were subjected to HPLC separation. For LC-MS experiments, the products were ionized in ESI mode to obtain their mass data (Table 4).

UPLC-MS/MS, UPLC-UV, HPLC-UV and UHPLC-UV methods were developed for multifaceted comparative analysis of TRELA stability (Table 4) [43]. A validated HPLC-UV method was selected for degradation kinetic study due to better separation for the degradation products and lower cost than the LC-MS/MS method. In the study of Luo et al. [44], rapid, sensitive and accurate LC-UV and LC-MS methods were developed for separation of TRELA and its eight potential process-related impurities. Between the methods, HPLC-UV using a XSelect CSH C18 column was proposed (Table 4).

It is worth noting that charged surface hybrid (CSH) technology can help overcome some problems such as peak shape asymmetry and poor loading for basic compounds, column bleed and slow mobile phase reequilibration. It was also found that the XSelect CSH C18 column yielded the best chromatographic separation between the critical pairs of peaks, while the other columns caused some peaks either to be co-eluted or to have comparatively low resolution. In addition, the XSelect CSH C18 column can give superior peak shape for basic compounds due to their improved loading capacity under low-ionic strength conditions and has excellent stability at low pH [50].

Forced degradation of VILDA was performed in extreme pH values and oxidative conditions [46]. Then, selective LC-UV was used for quantitative determination of a nondegraded drug in the presence of its degradation products. Finally, identification of degradation products of VILDA was performed using an UHPLC-DAD-MS with positive ESI (Table 4). The main objective of the next chromatographic method [47] was to achieve the separation of VILDA from VILDA-D4 and VILDA-D5 (Table 4). Additionally, a simple RP-HPLC-UV method was developed for separation of VILDA and its new degradation products, e.g., VILDA-D6, VILDA-D7, VILDA-D8 and VILDA-D9. Acetate buffer of pH 7.5 and MeOH with an Athena C18-WP (250 mm) column was used [48]. Athena C18-WP packing has great stability due to the high purity of spherical silica and can be used with 100% water as mobile phase. As a consequence, it can be used for the separation of many compounds with different acid-base properties. In the present study, phosphate or acetate buffers were mixed with MeOH in the ratio 90:10 (*v*/*v*). biomedicines-11-01956-t004_Table 4Table 4LC and LC-MS methods for determination of degradation products of gliptins: alogliptin (ALO), anagliptin (ANA), linagliptin (LINA), saxagliptin (SAXA), sitagliptin (SITA), teneliglitin (TENE), trelagliptin (TRELA) and vildagliptin (VILDA).CompoundConditionsRef.ALOGemini-NX C18 column (4.6 mm × 250 mm, 5 µm) and gradient elution with (A) 0.2% HCOOH-0.2% ammonium acetate and (B) ACN and MeOH (60:40, *v*/*v*) and PDA detection. ESI source of the Q-TOF unit in positive ionization mode.MS: spray voltage, 3.5 kV; N_2_ (drying gas) temperature, 350 °C with the nebulizer pressure of 207 kPa; fragmentor voltage, 175 V and collision energies 10–15 eV.[27]ALOUPLC-PDA: Kromasil C18 column (4.6 × 250 mm, 5 µm) and gradient with (A) 0.1% HClO_4_ in H_2_O (pH adjusted to 3.0 with TEA) and ACN in the ratio of 90:10 and (B) 0.1% HClO_4_ in H_2_O (pH adjusted to 3.0 with TEA) and ACN in the ratio of 40:60, 1 mL/min. UV 224 nm and column oven temperature 30 °C.LC-MS: Q-TOF mass spectrometer, (A) 0.1% HCOOH in H_2_O (pH adjusted to 3.0 with NH_4_OH) and ACN in the ratio of 90:10 and (B) 0.1% HCOOH in H_2_O (pH adjusted to 3.0 with NH_4_OH) and ACN in the ratio of 40:60. MS: positive and negative ESI, capillary voltage 4000 V, gas temperature 400 °C, fragmentor voltage, 125 V; skimmer voltage, 60 V and collision energy, 30 V.[28]ANAUPLC-PDA: Aquity BEH C18 column (2.1 × 100 mm, 1.7 µm) and gradient with (A) 10 mM ammonium formate without adjusting pH (measured pH 6.2) and (B) ACN (100% *v*/*v*), 0.3 mL/min; UV 248 nm. LC/MS: Q-TOF mass spectrometer, ESI positive mode: capillary voltage, 4.00 KV; fragmentor voltage, 170 V; skimmer voltage, 65 V; desolvation gas flow, 10 L/min; gas temperature, 325 °C, nebulizer gas, 40 psi.[29]ANAHPLC-PDA: a Daisogel-SP-100–10-ODS-P column (20 × 250 mm, 10 μm), 5 mL/min, UV 247 nm. LC-MS system with ion trap in positive ESI with ion source voltage of 5000 V and a source temperature of 450 °C; the nebulizer 20 psi using N2.[31]LINA in the presence of metforminHPLC-PDA: InertSustain^®^ C8 column (4.6 × 150 mm, 5 μm) and gradient elution with (A) 10 mM ammonium acetate (pH adjusted to 5.5 with CH_3_COOH) and (B) mixture of ACN and MeOH in the ratio of 80:20 (*v*/*v*), 0.9 mL/min; UV 254 nm. LC/Q-TOF: Acquity CSH C8 column (2.1 × 100 mm, 1.7 μm), 0.3 mL/min. Positive ESI: fragmentor voltage, 70 V; capillary voltage, 3500 V; skimmer voltage, 60 V; drying gas flow rate, 10 L/min; drying gas heater temperature, 320 °C; nebulizer gas pressure, 45 psi. N_2_ as drying, nebulizing and collision gas for CID.[32]LINALC-MS: Zorbax SB-Aq column (4.6 × 250 mm, 5 µm) and gradient elution with (A) phosphate buffer (0.02 M) of pH 3.0 and (B) ACN:MeOH (90:10, *v*/*v*), UV 225 nm. The column temperature 45 °C. UPLC-TOF/MS: Acquity BEH shield RP18 column (2.1 × 100 mm, 1.7 µm) and gradient elution with (A) ammonium acetate buffer (pH 3.8) and (B) ACN:MeOH (90:10, *v*/*v*), 0.21 mL/min. Positive ESI: capillary voltage at 3.0 KV, source and desolvation temperature 120 °C and 500 °C. The cone and desolvation gas flows 60 and 800 L/h.[34]SITALC-PDA: Zorbax Eclipse Plus C 18 column (4.6 × 100 mm, 5 µm) and gradient elution with (A) 10 mM ammonium formate and (B) MeOH, 0.5 mL/min.ESI-Q-TOF-MS/MS: positive ESI: spray voltage, 4.8 kV; CV, 20 V; capillary temperature, 300 °C and tube lens offset voltage, 10 V. N2 as a sheath gas (40 psi), and He as a damping and collision gas. CID experiments: collision energies between 15 and 25 eV.[37]SITALC-MS: RP18 column and gradient elution with A) ACN-H_2_O (1:99, *v*/*v*) with 10 mM ammonium formate (0.1%) and B) ACN-H_2_O (95:5, *v*/*v*) with 10 mM ammonium formate (0.1%), 0.4 mL/min. Positive ESI: N2_,_ flow rate 12 L/min, nebulizer pressure, 30 psi; gas temperature, 200 °C; sheath gas temperature, 350 °C; sheath gas flow, 12 mL/min; VCap, skimmer, 65 V; fragmentor voltage, 150 V; octopole RF Peak, 750 V and CID, 20 eV. [38]SITAUPLC-UV: Acquity BEH C-18 column (2.1 × 50 mm, 1.7 µm) and gradient program with (A) 10 mM ammonium formate, pH 6.4 and (B) CAN, 0.15 mL/min, UV 267 nm. Positive ESI: cone voltage, 20 V; CV, 3 KV; desolvation gas N_2,_ 1000 L/h, cone gas N2_,_ 50 L/h, source temperature, 180 °C; desolvation temperature, 500 °C. For fragmentation of selected ion collision energy 30 V. [39]SITALC-PDA: Gemini C18 110 A column (2 × 100 mm, 3 µm) and gradient elution with (A) ACN-H_2_O (1:99, *v*/*v*) with 10 mM ammonium formate (0.1%) and (B) ACN-H_2_O (95:5, *v*/*v*) with 10 mM ammonium formate (0.1%), 0.4 mL/min. LC-Q-TOF: positive ESI; N2 flow rate 12 L/min; nebulizer pressure, 30 psig; gas temperature, 200 °C; sheath gas temperature, 350 °C, sheath gas flow, 12 mL/min; VCap, 4000 V; skimmer, 65 V; fragmentor voltage, 150 V and octopole RF Peak, 750 V and CID, 20 eV.[40]TENEHPLC-UV: Kromasil^®^ 100-5 C18 (4.6 × 250 mm, 5 μm) and isocratic elution with pH 6.0 phosphate buffer and ACN (60:40 *v*/*v*), 1 mL/min.UPLC-PDA: Acquity BEH C18 column (2.1 × 50 mm, 1.7 μm) and gradient elution with A (10% ACN in H_2_O with 0.1% HCOOH) and B (90% ACN with 0.1% HCOOH), 0.3 mL/min.[41]TRELAUPLC-UV: BDS HYPERSIL C18 column (3 mm × 50 mm, 1.9 μm) and HYPERSIL Gold C18 column (3 mm × 50 mm, 1.9 μm), UHPLC-UV on SYMMETRY C18 column (2.1 mm × 100 mm, 2.2 μm) and isocratic elution with ACN-potassium dihydrogen pH 3.5 phosphate buffer 0.05 M (50:50, *v*/*v*).UPLC-MS/MS: Agilent SB-C18 column (2.1 × 50 mm 1.8 μm) and a mixture of ACN-HCOOH 0.1% (80:20, *v*/*v*); 120 °C, capillary voltage of 3 KV, dwell at 0.161 s, desolvation gas flow rate at 500 L/h, desolvation temperature at 400 °C.[43]TRELALC-UV: XSelect CSH C18 column (4.6 mm × 250 mm, 5 μm) and 0.05% TFA in H2O or ACN containing 0.05% TFA; UV 224 nm and 275 nm.UPLC-MS: BDS Hypersil C18 column (2.1 × 150 mm, 2.4 μm) and gradient elution with (A) 0.1% HCOOH in H_2_O and (B) ACN, 0.3 mL/min, column temperature 35 °C.UHPLC-LTQ-Orbitrap: ESI, the capillary temperature 250 °C, source voltage and spray voltage were 5 kV, sheath gas (N2) flow 35 psi.[44]VILDAUHPLC-PDA-MS: Kinetex XB-C18 column (2.1 × 150 mm, 1.7 µm) and gradient elution with (A) 0.1% HCOOH in H_2_O and (B) 0.1% HCOOH in ACN, 0.3 mL/min, column temperature 25 °C. MS: CP, 4500 V; endplate, 500 V, nebulizer pressure, 40 psi; drying gas temperature, 145 °C and gas flow rate, 9 L/min.[46]VILDAUPLC/Q-TOF: BEH C8 column (2.1 × 50 mm, 1.7 μm) and gradient with (A) H_2_O with 0.1% HCOOH and (B) MeOH containing 0.1% HCOOH, 0.3 mL/min, column temperature 35 °C.MS: positive ESI, CV 2.50 kV, cone voltage 30 V, extractor cone voltage 3 V, desolvation gas 400 L/h and cone gas 10 L/h; desolvation temperature 400 °C, source temperature 120 °C. [47]ACN—acetonitrile; TFA—trifluoroacetic acid.

## 3. Conclusions

All gliptins except for LINA predominantly undergo renal excretion, with ca. 70–80% of the dose eliminated as unchanged parent compound in urine. In contrast, LINA is excreted mostly unchanged in feces, and therefore appears to be safe in diabetic patients with renal disturbances. What is more, in patients with liver impairments, the dose adjustment seems unnecessary. However, the dose reduction of the gliptins with predominantly renal excretion (e.g., ALO, SITA, SAXA, but not VILDA) is needed in the case of such renal complications [7].

As far as metabolic transformations of gliptins are concerned, they undergo CYP-dependent and CYP-independent metabolism, and after that some of them have active metabolites. Bearing in mind all of the above results, it could be summarized that the dominant metabolic pathways for gliptins are hydroxylation and/or oxidation, affecting the butinyl side chain and particular heterocyclic moieties (Figure 2 and Figure 3).

As far as the reactions of II phase of the metabolism are concerned, N-acetylation (ALO, LINA), glucuronidation (EVO, SITA, VILDA) as well as sulfation (EVO, SITA) were reported, although these products, especially glucuronides, were further transformed and were not typically detected in excreta. After oral administration of VILDA, the cysteine- and glutathione-containing adducts were also detected in rat plasma, urine, feces and bile. What is more, formation of these metabolites was confirmed by in vitro incubation experiments. In addition, the possibility of forming the adducts with cysteine was also shown for ANA, EVO and LINA. It is worth noting that such reactivity may lead to unpredictable immune responses in humans [23].

It was interesting to observe that metabolism and drug degradation pathways may overlap for many gliptins (ANA, ALO, SAXA, TRELA and VILDA), resulting in the formation of similar constituents. The major transformation of the above drugs, during metabolism as well as degradation, was proposed as the cyano group hydrolysis to generate the carboxylic acid or amide, resulting in such products as ANA-M1 and VILDA-M1 (Figure 4).

At the same time, similar reactions were confirmed while stress degradation in different conditions was performed, leading to generate ALO-D3, ANA-D3 and ANA-D-4, SAXA-D2 and SAXA-D-4, TRELA-D2 and TRELA-D-4, and VILDA-D2 and VILDA-D-4. The presented results pointed to hydrolysis of the cyano group in basic, oxidative, as well as acidic conditions. Further hydrolysis of the amide group afforded the corresponding carboxylic acid.

Metabolism data could be helpful for deriving the safe levels of the related compounds by using it to qualify these impurities to reduce the cost, animal use and time. Thus, our considerations about gliptins could be leveraged to enhance their behavior in pharmaceutical products and link the quality of respective pharmaceutical products to clinical practice. As was also shown, the main analytical tools for such modern analysis of gliptins are chromatographic methods like LC-MS or HRMS. Such hyphenated techniques present the power of separation and quantification for more in-depth analysis of complex samples from metabolism and degradation experiments, in turn allowing us to solve these complex problems. In addition, radiometric methods using the stable isotopes, as well as chromatographic methods with the deuterated standards, could be used for determination of the mentioned drugs in biological fluids or excreta. As was described above, [14C]-technology was widely applied to track the gliptins and to provide their metabolic profiles.

## Figures and Tables

**Figure 1 biomedicines-11-01956-f001:**
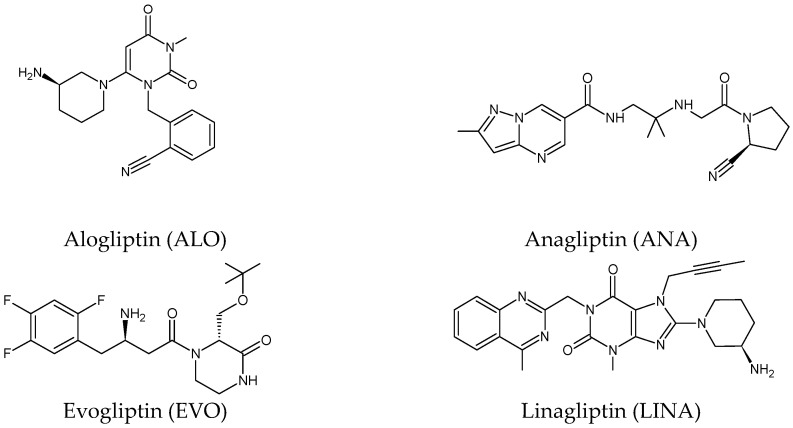
Chemical structures of DPP-4 inhibitors: alogliptin (ALO), anagliptin (ANA), evogliptin (EVO), linagliptin (LINA), omarigliptin (OMA), saxagliptin (SAXA), sitagliptin (SITA), teneligliptin (TENE), trelagliptin (TRELA) and vildagliptin (VILDA).

**Figure 2 biomedicines-11-01956-f002:**
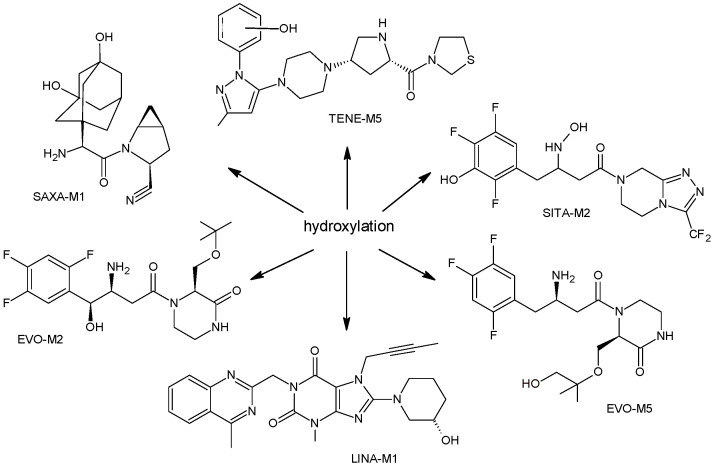
The metabolic transformations of DPP-4 inhibitors via hydroxylation: metabolites of evogliptin (EVO-M2 and EVO-M5), linagliptin (LINA-M1), saxagliptin (SAXA-M1), sitagliptin (SITA-M2) and teneligliptin (TENE-M5).

**Figure 3 biomedicines-11-01956-f003:**
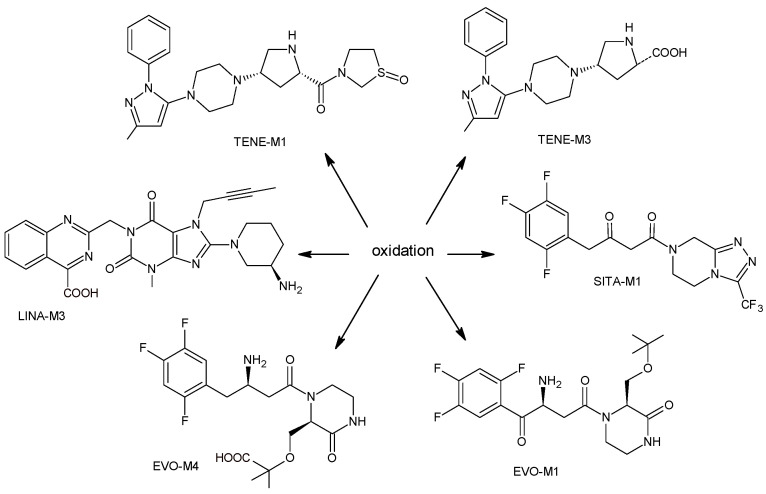
Metabolic transformations of DPP-4 inhibitors via oxidative transformations: metabolites of evogliptin (EVO-M1 and EVO-M4), linagliptin (LINA-M3), sitagliptin (SITA-M1) and teneligliptin (TENE-M1 and TENE-M3).

**Figure 4 biomedicines-11-01956-f004:**
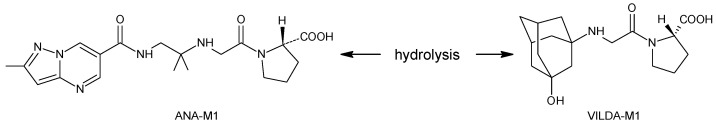
Metabolic transformation of anagliptin (ANA) and vildagliptin (VILDA) via hydrolysis of the cyano group.

## Data Availability

All data relevant to this study are included in the manuscript.

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
