# Peer review of "Metabolism and Chemical Degradation of New Antidiabetic Drugs (Part II): A Review of Analytical Approaches for Analysis of Gliptins"

_biomedicines, 2023, doi:10.3390/biomedicines11071956_

Round 1

Reviewer 1 Report

Dear authors,

Thank you for compiling this review. As such you have done tremendous work however it is not easy to follow. The flow of the paper is missing and the content is hopping in many directions which makes it difficult to understand. I would suggest you organise all the drug metabolism and stability data in the beginning and then deal with the analytical methodology later so that readers find it easy to understand. If possible please cut down on so many chemical structures or organise them based on drug by drug basis, ie drug followed by their metabolism/stability studies and analytical method if you do not want to deal with them in end rather than discussing them all together. 

Best

Please take the help of an experienced English writer or software (Grammarly), as some of the sentences are not easy to understand.

Author Response

Reviewer 1

Thank you for compiling this review. As such you have done tremendous work however it is not easy to follow. The flow of the paper is missing and the content is hopping in many directions which makes it difficult to understand. I would suggest you organise all the drug metabolism and stability data in the beginning and then deal with the analytical methodology later so that readers find it easy to understand. If possible please cut down on so many chemical structures or organise them based on drug by drug basis, ie drug followed by their metabolism/stability studies and analytical method if you do not want to deal with them in end rather than discussing them all together.

Answer:

Thank you very much for your positive opinion about our article and for all the Reviewer's comments. Indeed, sometimes our text is difficult to read due to the amount of information we have included in it. In the corrected version an attempt was made to follow the Reviewer's suggestion to discuss each drug separately, but we failed to achieve greater consistency and to reduce the number of structures. Therefore, we improved our article in such a way as to separate the parts of the material related to particular analytical procedures. Therefore, there are two additional subsections now, i.e., 2.1.1. and 2.1.2. The text of the entire article was revised to make it clearer and easier for the reader to follow. We hope that the current version will find appreciation in the eyes of the Reviewer.

Reviewer 2 Report

Metabolism and chemical degradation of new antidiabetic drugs (Part II): a review of analytical approaches for analysis of gliptins

Gumieniczek, A.; Berecka-Rycerz, A.

The manuscript entitled “Metabolism and chemical degradation of new antidiabetic drugs (Part II): a review of analytical approaches for analysis of gliptins” by Gumieniczek & Berecka-Rycerz shows a significant amount of academic research done in the field of metabolism and chemical degradation of antidiabetic drugs.

The manuscript’s abstract is promising since it reflects on the overall reason for monitoring drug metabolism, degradation, and excretion, i.e., avoiding toxic degradation impurities and deriving safe doses (although the latter is not stated in detail). The review itself goes in-depth into the different metabolites of DPP-4 inhibitors after Phase I and Phase II reactions and the analytical methods to analyze them, such as liquid chromatography (LC), liquid chromatography coupled with mass spectrometry (LC-MS or LC-MS/MS), radiometric methods (radioactivity detector, accelerator mass spectrometry), etc. 

The authors also present relevant clinical data, such as the analytical data obtained, when a combined therapy (DPP-4 inhibitors + metformin) is given to Diabetes type-2 patients.

I have just one minor observation:

Line 314. TFA-trifluoroacetic acid. This phrase is not connected to the main text or Table 2. Please correct it.

Author Response

Reviewer 2

The manuscript entitled “Metabolism and chemical degradation of new antidiabetic drugs (Part II): a review of analytical approaches for analysis of gliptins” by Gumieniczek & Berecka-Rycerz shows a significant amount of academic research done in the field of metabolism and chemical degradation of antidiabetic drugs.

The manuscript’s abstract is promising since it reflects on the overall reason for monitoring drug metabolism, degradation, and excretion, i.e., avoiding toxic degradation impurities and deriving safe doses (although the latter is not stated in detail). The review itself goes in-depth into the different metabolites of DPP-4 inhibitors after Phase I and Phase II reactions and the analytical methods to analyze them, such as liquid chromatography (LC), liquid chromatography coupled with mass spectrometry (LC-MS or LC-MS/MS), radiometric methods (radioactivity detector, accelerator mass spectrometry), etc. 

The authors also present relevant clinical data, such as the analytical data obtained, when a combined therapy (DPP-4 inhibitors + metformin) is given to Diabetes type-2 patients.

I have just one minor observation:

Line 314. TFA-trifluoroacetic acid. This phrase is not connected to the main text or Table 2. Please correct it.

Answer

Thank you very much for this positive opinion about our article and for all Reviewer's comments.

Line 314 was corrected in our revised text.

Also, one additive sentence about the efforts to avoide toxic degradation impurities and derive safe doses of the drugs was added in our Conclusions on page 25. The detailed considerations on the toxicological assessment and qualification of impurities were beyond the scope of our paper. However, thank you very much for this comment and suggestion to write a new paper which would cover these interesting issues.

Reviewer 3 Report

Dear Authors,

In the manuscript “Metabolism and chemical degradation of new antidiabetic drugs (Part II): a review of analytical approaches for analysis of gliptins”, the second part of the review on metabolism and chemical degradation of gliptins (dipeptidyl peptidase 4 (DPP-4) inhibitors, antidiabetic drugs) is presented. Analytical tools used to monitor drug metabolism and drug degradation were summarized. These methods include LC-UV, LC-MS and LC-MS/MS that are widely used for detection and quantitative measurements of the drugs, their metabolites and degradants. Radiometric methods were also included. Chromatographic conditions (columns, gradients) along with MS parameters were tabulated.

Reviewer’s notes. 1. Line 48. Should be “finally”.

2. Line 65. Should be “process”, double “s”).

3. Page 4, top. Figure 1. Alogliptin: what configuration has C atom bearing amino group? (The same as in trelagliptin.)

4. Lines 160-162. Table 1. Some m/z values are not shown. Same for Table 3 (lines 366-368).

5. Lines 243-245. The sense of this phrase is unclear: what is the scheme of the used instrument?

6. Line 311 and below, Table 2. What is Arb? Arbitrarily units?

7. Lines 464-465. The sense of this phrase is unclear: is it the same product or isomers?

8. Line 656. Please specify conflict of interests (or its absence).

English is quite good though moderate language editing is desirable.

Author Response

Reviewer 3

In the manuscript “Metabolism and chemical degradation of new antidiabetic drugs (Part II): a review of analytical approaches for analysis of gliptins”, the second part of the review on metabolism and chemical degradation of gliptins (dipeptidyl peptidase 4 (DPP-4) inhibitors, antidiabetic drugs) is presented. Analytical tools used to monitor drug metabolism and drug degradation were summarized. These methods include LC-UV, LC-MS and LC-MS/MS that are widely used for detection and quantitative measurements of the drugs, their metabolites and degradants. Radiometric methods were also included. Chromatographic conditions (columns, gradients) along with MS parameters were tabulated.

Thank you very much for this positive opinion about our article and for all the comments.

  1. Line 48 and 65:

Answer

Lines 48 and 65 were corrected in our revised text.

  1. Page 4, top. Figure 1. Alogliptin: what configuration has C atom bearing amino group? (The same as in trelagliptin.)

Answer

Thank you very much for this comment. The structure of Alogliptin was corrected.

  1. Lines 160-162. Table 1. Some m/z values are not shown. Same for Table 3 (lines 366-368).

Answer

Thank you very much for this comment. Respective m/z values were added, wherever respective data was found in the literature.

  1. Lines 243-245. The sense of this phrase is unclear: what is the scheme of the used instrument?

Answer

The text of the entire article was revised to make it clearer and easier for the reader to understand. We hope that the current version will find appreciation in the eyes of the Reviewer.

  1. Line 311 and below, Table 2. What is Arb? Arbitrarily units?

Answer

Yes, such information was added under respective Table.

  1. Lines 464-465. The sense of this phrase is unclear: is it the same product or isomers?

Answer

The text of the entire article was revised to make it clearer and easier for the reader to understand. We hope that the current version will find appreciation in the eyes of the Reviewer.

  1. Line 656. Please specify conflict of interests (or its absence).

Answer

Thank you very much for this comment. It was corrected in our revised text.

Round 2

Reviewer 1 Report

Dear Authors,

Thanks very much for revising the manuscript. One last suggestion, please change all m/z to the same significant figures in Tables 1 and 3. Indicate if this measurement is from an HRMS instrument for example ALO-D1 m/z = 258.0873. and  ANA-D1 m/z= 225 ?? is from a low-resolution mass spec if that is the case.

Best wishes,

**

Author Response

Dear Authors,

Thanks very much for revising the manuscript. One last suggestion, please change all m/z to the same significant figures in Tables 1 and 3. Indicate if this measurement is from an HRMS instrument for example ALO-D1 m/z = 258.0873. and  ANA-D1 m/z= 225 ?? is from a low-resolution mass spec if that is the case.

Answer

Thank you for this suggestion. The m/z values are unified now, and those obtained with low resolution mass spectrometry (with maximum 2 decimal places) are indicated by * while HRMS data are marked by **.